# Sleep Instability Correlates with Attentional Impairment in Boys with Attention Deficit Hyperactivity Disorder

**DOI:** 10.3390/brainsci11111425

**Published:** 2021-10-27

**Authors:** Katia Gagnon, Mélanie Labrosse, Marc-André Gingras, Roger Godbout

**Affiliations:** 1Sleep Laboratory and Clinic, Hôpital en Santé mentale Rivière-des-Prairies, Montréal, QC H1E 1A4, Canada; katia.gagnon@umontreal.ca (K.G.); melanie.labrosse@umontreal.ca (M.L.); magingras.neuropsychologue@gmail.com (M.-A.G.); 2Department of Psychiatry, Université de Montréal, Montréal, QC H3T 1J4, Canada

**Keywords:** attention deficit and hyperactivity disorder, polysomnography, sleep instability, stage shifting, cognition, attention functioning

## Abstract

Theoretical models of sleep and attention deficit hyperactivity disorder (ADHD) suggest that symptoms of ADHD are associated with daytime sleepiness, but it has received little support. The present study aimed at testing an alternative model involving the association of attentional instability with sleep instability, i.e., sleep stage transitions and arousals. Twelve ADHD and 15 healthy control (HC) boys aged between 8 and 12 years old underwent polysomnography recording and attentional testing. The microarousal index, the number of awakenings, and the number of stage shifts between stages 1, 2, 3, 4 and REM sleep throughout the night were computed as sleep stability parameters. Attentional functioning was assessed using the Continuous Performance Test-II. We found significantly higher sleep instability in ADHD compared to HC. Sleep arousals and stage transitions (micro arousal index, stage 4/3 and 2/4 transitions) in ADHD significantly correlated with lower attentional scores. No association whatsoever was found between sleep instability and attentional functioning in HC. The results show that sleep instability is associated with lower attentional performance in boys with ADHD, but not in HC. This could be compatible with a model according to which attention and sleep stability share a common neural substrate in ADHD.

## 1. Introduction

Attention deficit hyperactivity disorder (ADHD), one of the most prevalent neurocognitive disorders in childhood, is characterized by impaired attention, impulsivity, and excessive motor activity, which reduces the quality of social, academic, or occupational functioning [1]. Although ADHD is associated with structural and functional changes in the brain, the neural mechanisms implicated in inattention and impulsivity are still unclear [2,3]. Several theoretical models of ADHD have been put forward, but they have often ignored sleep issues that occur in almost half of children with ADHD, based mostly on parental reports [4,5], while sleep macrostructure as measured with polysomnography appeared to be less sensitive [4,6,7].

Over the past years, several models linking ADHD signs and symptoms to sleep have been proposed. Since neurochemical and anatomical structures that modulate attention are also involved in arousal and sleep, a consensus working group of researchers suggested that sleep and attention dysfunction in children with ADHD could be an expression of the alteration of the neural circuitry that regulates both sleep/wake and attention [6]. Acting as a common pathophysiological pathway, defects in neuronal networks in ADHD could thus lead to both sleep instability, defined here as arousals/microarousals and sleep stage shifts, as well as impaired cognitive performance.

Very few studies have investigated sleep stability in ADHD, yielding mixed results, possibly because varied methods were used [7,8,9]. Moreover, these studies have not tested the link between sleep instability and cognitive functioning in ADHD. Our study aimed at comparing sleep stability of children with ADHD to healthy control (HC) children using polysomnography, paired with cognitive performance testing. We hypothesized that: (a) sleep macrostructure would not be different between ADHD children and HC children; (b) ADHD children would display more sleep instability compared to HC children; and (c) sleep instability would be associated with a lower performance on cognitive measures of attention in children with ADHD.

## 2. Materials and Methods

### 2.1. Participants

Twelve right-handed boys with ADHD, aged between 8 and 12 years old, were recruited through a specialized clinic for ADHD at the Hôpital en santé mentale Rivière-des-Prairies, (Montréal, QC, Canada). Inclusion criteria were: (1) a diagnosis of ADHD made by a psychiatrist; (2) no comorbid psychiatric, neurological, or medical disorders, including sleep disorders, as determined by a sleep specialist on site; and (3) a full-scale intelligence quotient (IQ) above 80, as measured by the Kaufman Brief Intelligence Test [10]. DSM-IV criteria were used to diagnose ADHD and other comorbid psychiatric disorders. Children with ADHD all met criteria for ADHD combined type [11]. All ADHD children were treated with the stimulant medication methylphenidate, except for two who were treated with atomoxetine, a non-stimulant medication. The ADHD group was compared to 15 right-handed (HC) boys, aged between 8 and 12 years old, recruited through public announcements. Exclusion criteria for HC boys were: (1) a personal history of psychiatric, neurological, or other medical conditions, including sleep disorders; (2) medication use that can affect the central nervous system; and (3) a full-scale IQ below 80 as measured by the Kaufman Brief Intelligence Test [10]. The ADHD and HC children were recruited and studied all year round, alternating between ADHD and HC participants most of the time in order to avoid a sequence effect.

The parents of all children completed the Child Behavior Checklist/4-18 (CBCL) [12]. Raw scores were converted into age-standardized scores. The total scores, the attention scale scores, and the anxiety/depression scores were used to compare ADHD and HC groups.

### 2.2. Procedure

Under the supervision of their parents, participants were asked to keep a regular sleep-wake schedule and complete a sleep diary for 14 days before coming to the laboratory. Parents were asked to complete a sleep questionnaire (Children’s Sleep Habits Questionnaire: CSHQ) [13] and an ADHD symptom severity questionnaire (Child Behavior Checklist/4-18) [12]. Napping, drinking caffeine beverages, and eating chocolate was not permitted on the day prior the recordings; none of the children were usual nappers. ADHD participants were withdrawn from psychostimulant medication (methylphenidate) for at least 48 h before sleep recording until the end of the study. Non-stimulant medication (atomoxetine, in two children) was maintained during the protocol. Participants slept in the laboratory for two consecutive nights, and their parents were sleeping in an adjoining bedroom. The first laboratory night served to screen sleep disorders, and to allow adaptation to the experimental environment. Polysomnographic data presented are from the second night. Cognitive testing was carried out between 8:00 a.m. and noon after night 2, in a quiet testing room, by a well-trained and experienced psychotechnician. 

The protocol was approved by the ethic committee of the Hôpital en santé mentale Rivière-des-Prairies (reference approval number 03-07). Informed consent was obtained from both children and parents. All participants received a financial compensation for their involvement in this research.

### 2.3. Polysomnographic Recordings

Sleep was recorded and scored according to standard methods, using 20-s epochs [14,15]. Electro-oculogram (EOG) electrodes were applied over each outer canthus, and surface electromyogram (EMG) electrodes applied over the submental muscles. Electroencephalographic (EEG) electrodes were positioned according to the international 10–20 system for frontal (F3, F4), central (C3, C4), and occipital (O1, O2) recording sites. EEG leads were referenced to linked earlobes (A1 + A2), with a serial 10 KΩ resistor for impedance equilibrium purposes [16,17]. Periodic leg movements in sleep (PLMS) were monitored with bilateral anterior tibialis muscle surface EMG. PLMS scoring criteria were as follows: at least 4 anterior tibialis EMG bursts separated by at least 5 s, and at most by 90 s intervals. Respiratory flow and effort were monitored using oronasal cannula and thoraco-abdominal strain gauges, respectively. Arterial oxygen saturation (SaO2) was measured with an infrared oximetry sensor to the index finger (Datex Ohmeda 3900 Pulse Oxymeter, Datex-Ohmeda, Louisville, CO, USA). The identification criteria for sleep apnea were those of the American Academy of Sleep Medicine [18].

A Grass Neurodata Model 12 Acquisition system was used for recording, and signals were digitized using Harmonie 5.0 Software (Stellate, Montreal, QC, Canada). Filter settings and amplification factors for EEG were: 1/2 amplitude high pass = 0.3 Hz, 1/2 amplitude low pass = 30 Hz, amplification × 1000 = 20; EOG: 1/2 amplitude high pass = 0.1 Hz, 1/2 amplitude low pass = 30 Hz, amplification × 1000 = 20; EMG: 1/2 amplitude high pass = 10.0 Hz, 1/2 amplitude low pass = 100 Hz, amplification × 300 = 20. Signals were sampled at 256 Hz. Records were stored for offline visual analysis on a computer screen.

### 2.4. Sleep Architecture

Sleep onset was defined as 10 consecutive minutes of stage 1, or the first epoch of any other sleep stage. Sleep latency to sleep stages was the interval between sleep onset and the first epoch of that stage. REM latency was the interval between sleep latency and the first REM sleep epoch. The sleep period was defined as the time elapsed from sleep onset to sleep offset (last awakening). Total sleep time was the total number of minutes spent during the sleep period. Sleep efficiency was computed by the following formula: total sleep stages duration/sleep period ×100; a REM sleep period was defined as a succession of REM sleep epochs not interrupted for more than 15 min. REM sleep efficiency was calculated as: total duration of REM sleep/total duration of REM periods × 100.

### 2.5. Sleep Stability Parameters

Sleep stability was assessed with three measures computed through the sleep period: stage shifts, awakenings, and microarousals. Stage shifts were identified as transitions from a sleep stage to wake or another sleep stage. Awakenings were defined as epochs scored as wake stage. Microarousals were defined as awakenings shorter than 10 s. The microarousal index corresponds to the number of events per hour of sleep.

### 2.6. Cognitive Testing

Attentional functioning was tested using the Continuous performance Test- Second edition (CPT-II), a computer-based program that measures attention problems in individuals of 6 years of age and older [19]. Letters were presented on the center of the screen, and children were required to press the spacebar as quickly as possible in response to any letters other than “X”, and not to respond when the “X” letter was presented. Letters were presented on the screen for 250 ms at varying time intervals of one-, two-, and four-second intervals between stimuli. There was a two-minute practice test, and then the main test was administered for an approximate duration of 14 min. The assessment was segmented into six blocks, each with three sub-blocks. A total of 360 stimuli, including 36 “X”, appeared over the duration of the test. Thirteen different metrics were calculated by the program, and converted into standard scores. Mean reaction time, omission errors (not responding to a letter other than “X”), and commission errors (responding to an “X”) were computed as measures of attention functioning. Reliability of these measures is estimated to be between 0.83 to 0.95 [19]. Standard scores were used for group comparison and correlations. Higher scores are associated with lower performance.

### 2.7. Questionnaires

#### 2.7.1. Child Behavior Checklist/4-18 (CBCL)

The Child Behavior Checklist/4-18 is a parent-completed questionnaire that measures children’s adaptive behavior using a three-level scale: “true or often true”, “somewhat or sometimes true”, and “not true” [12]. The scale yields eight behavioral constructs, including anxious/depressed and attention problems. The CBCL generates a total score that can be converted into standardized scores. Higher scores are associated with more behavioral issues, and a T score above 70 is in the clinical range. CBCL is a valid and clinically useful instrument for the evaluation of children referred for ADHD, and the attention problem subscale is a good predictor of ADHD diagnosis [20,21,22,23].

#### 2.7.2. Children’s Sleep Habits Questionnaire (CSHQ)

The Children’s Sleep Habits Questionnaire (CSHQ) is a retrospective, 45-items questionnaire that was developed to examine sleep behavior in children [13]. Parents were asked to rate their children sleep behaviors over a typical week. The CSHQ uses three level scales: “usually” (5 to 7 times per week), “sometimes” (2 to 4 times per week), and “rarely” (0 to 1 time per week). The questionnaire is segmented into subscales, including daytime sleepiness. A higher score indicates more disrupted sleep. Test-retest reliability show a Cronbach’s alpha of 0.78 for global scale, and a range between 0.56 and 0.93 for subscales [24]. A score of 41 or above is in the clinical range. The total score showed good validity to differentiate controls from a sleep-disorder group (sensitivity 0.80 and specificity of 0.72; ROC curve = 0.41) [13].

### 2.8. Statistical Analysis

Statistical analyses were performed with SPSS 25.0 (SPSS Science, Chicago, IL, USA), and statistical significance was set at *p* < 0.05. Distribution normality was investigated, and appropriate transformations were applied to non-normally distributed variables. Power analysis, based on the literature on sleep instability in ADHD children, shows that a sample of 26 participants can provide large effect sizes (α = 0.05; 1 − β = 0.80).

Because there are no clinical cutoff scores for the CSHQ sleepiness subscale, we transformed the individuals’ absolute scores into Z scores, using the mean and standard deviations of the 381 healthy control children in Owens et al., 2000, and we used two standard deviations over the mean as a cutoff score [13].

Student t-tests were used to test group differences (ADHD vs. HC) on demographic, sleep, and cognitive variables. Levene’s tests were applied for homogeneity of variance. For each group, we used Pearson’s correlations between sleep stability variables that were found to be significantly different between groups, and cognitive measures of attention on CPT-II (reaction times, omission errors, commission errors) to test the relation between sleep stability and cognitive functioning. Pearson’s correlations were also used to measure the association between sleep stability variables and CBCL attention scores.

## 3. Results

### 3.1. Clinical Characteristics and Sleep Macroarchitecture

Table 1 shows that groups were not different on age, body mass index, and IQ. Compared to HC, the ADHD group displayed significantly higher scores on the CBCL total and attention scores, but no significant differences were found on the anxiety/depression scores. No significant differences were found on the CPT-II. The total score on the CHSQ was higher in the ADHD group compared to HC, but no differences were found on the sleepiness subscale. Table 2 shows that ADHD and HC groups were not different on sleep macroarchitecture variables. Levene’s tests were not significant.

### 3.2. Sleep Stability

Compared to HC, the ADHD group showed a higher microarousal index during Stage 1, as well as more transitions from Stage 2 to REM sleep, and from REM sleep to Stage 2; transitions from Stage 4 to Stage 3 were also higher in the ADHD group compared to HC. The HC group displayed a higher number of transitions from Stage 2 to Stage 4, and from Stage 4 to Stage 2 compared to the ADHD group (Table 3). Levene’s tests were not significant.

### 3.3. Association between Sleep Stability and Attention

In the ADHD group, higher omission scores (*r* = 0.42; *p* = 0.048) and commission scores (*r* = 0.82; *p* = 0.002) were significantly correlated with higher Stage 1 microarousal index values (Figure 1A,B) while higher omission scores were associated with more transitions from Stage 4 to Stage 3 (*r* = 0.46; *p* = 0.03) (Figure 1C). There were no significant correlations CPT-II scores and REM sleep stability variables. No significant correlations between sleep stability variables and CPT-II measures whatsoever were found in the HC group. No significant correlations were found between CBCL attention scores and sleep stability in either of the two groups.

## 4. Discussion

As hypothesized, the ADHD group showed a higher number of transitions from deeper to lighter stages of sleep, compared to the HC group. They also showed a higher number of bidirectional transitions between REM sleep and Stage 2. Interestingly, only transitions to lighter non-REM sleep stages positively correlated with lower scores on objective attentional measures in the ADHD group. The HC group displayed significantly more bidirectional transitions between sleep Stages 2 and 4, but these measures did not correlate with attentional performance. Taken together, these results suggest that sleep instability towards lighter sleep stages of non-REM sleep and lower attentional performance share a common pathophysiological pathway in children with ADHD that is not active in HC.

### 4.1. How the Present Results Fit into Existing Models of Sleep Instability in ADHD?

There are three major models of sleep instability in ADHD. The first model suggests that ADHD could be associated with an hypoarousal state resembling narcolepsy [25]. This model involves an increase in daytime sleepiness, and a decrease in the markers of sleep stability, as measured by electroencephalography (EEG) cycling alternating patterns [7,25]. Two studies found a lower rate of cycling alternating patterns in ADHD children [7,8], while a third one did not [9]. More importantly, one of these studies did not assess sleepiness [9], while the two others, either using their own children-adapted version of the Epworth Sleepiness Scale or the Sleep Disturbance Scale for Children [26], found sleepiness in only two of 20 [7] and three of 28 [8] ADHD children, respectively. The present study found more sleep instability in boys with ADHD compared to controls, but we did not find between-group differences on either subjective (CSHQ) or objective (CPT-II) measures of sleepiness. Moreover, only one ADHD child in the present study had a score on the CHSQ’s sleepiness subscale that exceeded 2 standard deviations over the mean of the 381 healthy control children, published in Owens et al., 2000 [13]. These results are not compatible with this first model.

A second model was recently proposed by Andrillion et al., 2019, according to which sleep and wakefulness are considered as a continuum, rather than mutually exclusive states [3]. Partially supporting this idea, studies using topographical analyses of the EEG during non-REM have documented local changes in sleep depth across the scalp during non-REM sleep in healthy adults without sleep disturbances [27,28]. Other studies have reported local sleep-like non-REM sleep slow waves occurring at the level of single-neuron activity during wakefulness and REM sleep [29,30,31,32,33,34]. Local sleep occurring during wakefulness has been associated with attentional lapses, slower reaction times, and increased error rates [28,29,30,32,34]. According to this model, the alteration of the neural circuits regulating attention and sleep/wake in ADHD could lead to a higher number of awakenings during sleep, or sleep instability, which could negatively impact daytime cognitive performance due to of daytime sleepiness [3]. Although we found a significant correlation between sleep instability and attentional functioning, the lack of sleepiness in our ADHD group may not support this model.

A third model was proposed by a multidisciplinary workgroup of pediatric sleep researchers, putting forward the idea that the neural circuitry involved in sleep/wake regulation also controls attention. Our results match well with this theoretical framework, as sleep instability, as measured by more microarousals and more transitions to lighter sleep stages, was associated with lower attentional functioning, without significant daytime sleepiness. Moreover, a study by Miano et al. also found a higher number of transitions between unspecified sleep stages in ADHD children compared to controls and, as in the present study, with only 2 of 20 ADHD children presenting symptoms of daytime sleepiness [7]. The association between sleep instability in ADHD and attentional measures, however, was not tested. Although the authors did not report on the direction in which sleep stage transitions occurred, these results suggest that neurodevelopmental defects associated with ADHD could have a negative impact on both sleep stability and attention functioning, but not on daytime sleepiness. The next section will discuss the potential neural structures and neurotransmitters that could be involved in both sleep instability and attentional performance in ADHD.

### 4.2. What Are the Potential Brain Networks Involved in Both Attention Functioning and Sleep Instability in ADHD?

It is known that individuals with ADHD exhibit alertness/attention instability [35]. Methylphenidate, a psychostimulant acting on the dopaminergic and noradrenergic systems, is commonly used to treat ADHD by improving attention stability [36,37]. The main neuroanatomical structure providing noradrenergic release throughout the brain is the locus coeruleus [38]. In addition to contributing to cognitive functions regulations, such as attention, vigilance, and memory, the locus coeruleus is also involved in sleep regulation, including sleep stage stability [39]. Over the following sections we will look at each significant sleep transitions in ADHD, and evaluate how the previous structures and neurotransmitters are potentially participating to sleep instability.

#### 4.2.1. Sleep Stage 1 to Micro-Arousal Oscillations

Microarousals are generated, at least partially, by the same structures involved in maintaining arousal [40]. The cerebral cortex, basal forebrain, and hypothalamus are diffusely innervated by monoaminergic groups of cells that drive arousal through the secretion of norepinephrine, serotonin, dopamine, and histamine [41,42]. The locus coeruleus, which is the largest group of noradrenergic neurons, plays a major key role in wakefulness through multiple excitatory projections to wakefulness-promoting nucleus, and inhibitory projections to sleep-promoting nucleus [41,43,44]. Thus, a neurodevelopmental impairment of the noradrenergic system could reduce alertness during the daytime, and increase sleep awakenings at night [43].

#### 4.2.2. Oscillations between Sleep Stages 3 and 4

The noradrenergic system, including the locus coeruleus, modulates cortical activity during non-REM sleep. Using a coupling method between EEG and functional magnetic resonance imaging, Dang-Vu et al., 2008 found a temporal relationship between locus coeruleus activity and slow EEG oscillations [44]. Eschenko et al., 2011 also showed that locus coeruleus firing was associated with the rising phase of the EEG slow waves in rodents [45]. The locus coeruleus is thus involved in neuromodulatory processes, by increasing cortical excitability [45]. As mentioned earlier, noradrenergic neurotransmission appears to be unstable in children with ADHD. Thus, the capacity to increase cortical excitability, or to trigger firing in the locus coeruleus in order to generate slow waves, could be reduced or unstable. As a result, the density of EEG slow waves might be more variable throughout the night, leading to more oscillations between Stages 3 and 4.

#### 4.2.3. Oscillations between REM and NREM Sleep

Pedunculopontine and laterodorsal tegmental cholinergic nuclei are involved in the transitions from non-REM to REM sleep [46]. The noradrenergic neurons of the locus coeruleus are known to have an inhibitory influence on REM sleep [39]. They stabilize non-REM sleep by inhibiting the cholinergic neurons from the pedunculopontine and laterodorsal tegmental [44,47,48]. Moreover, increasing noradrenaline levels in the brain using antidepressants can suppress REM sleep [49]. In this context, unstable noradrenergic system could lead to more transition between REM and non-REM sleep.

### 4.3. What about the Brain Maturation Delay Hypothesis and Sleep Instability in ADHD?

The maturation delay hypothesis, or maturational lag model, was proposed in 1973 by Kinsbourne [50]. This model suggests that there is a developmental lag in the central nervous system, which leads to a delayed maturation of cognitive processes in children with ADHD; age-related cognitive performance impairment of ADHD children could correspond to a normal performance at a younger age. The ADHD maturational lag model is largely supported by EEG findings, as well as neuropsychological and neuroanatomical studies [47,48,51,52,53]. Like cognitive maturation, healthy sleep follows a neurodevelopmental process, which can be monitored through EEG. Interestingly, sleep instability found in ADHD seems to fit well into the maturational lag model. The next paragraph will discuss the ontogenetic development of sleep stability for each of the significant unstable oscillations that we found in our sample of ADHD children.

Microarousals are a natural component part of sleep in healthy sleepers [54]. Studies in rodents and humans showed that the number of microarousals decreases during brain maturation [54,55]. Thus, the higher number of microarousals that we found in children with ADHD could represent a brain maturational delay. In the same vein, REM/non-REM sleep cycle oscillations are short in infants, and tend to lengthen with brain maturation [41]. Consequently, an increasing number of short transitions between non-REM and REM sleep could also be associated with a brain maturation delay. Finally, oscillations between Stages 3 and 4 are based on the density of EEG slow waves in sleep recording samples. A density of 20% to 49% of a sample is considered as Stage 3, while 50% and more is considered as Stage 4 [56]. Healthy brain maturation during childhood is associated with high levels of EEG slow waves until adolescence, where a steady decrease is initiated [57,58]. In this context, a neurodevelopmental delay could lead to lower EEG slow wave density during childhood, and be associated with more oscillations between Stages 3 and 4.

According to the model, sleep instability in ADHD could be attributable to a maturational brain delay. Thus, sleep disturbances and EEG markers found in ADHD could correspond to the sleep of a younger child, but this hypothesis needs to be tested exhaustively.

### 4.4. Strengths and Limitations

This innovative study is the first to analyze the association between sleep stability and objective attentional performance in children with ADHD, taking sleepiness into consideration. On the one hand, the small number of participants calls for larger confirmatory studies. On the other hand, participants were carefully selected to reduce the probability of type 1 errors. The fact that only boys were studied controlled the variability associated with sex differences in the presentation of ADHD [58]. All of the boys had an ADHD diagnosis without any psychiatric comorbidity nor primary sleep disorders, and psychostimulant medication was withdrawn for 48 h before entering the study. Another strength is that a habituation night allowed children to get used to the laboratory setting before data was collected. One-to-one matching by age, body mass index, and psychometric measures in two larger groups of participants would improve the generalizability of the present results.

## 5. Conclusions

In recent years, models of sleep and ADHD have included daytime sleepiness as a moderator for the association between sleep disturbances and attentional performance in ADHD. Although we found clinically significant sleep disturbances in ADHD according to parental reports the CSHQ, daytime sleepiness was not significant in this group. Previous studies did not demonstrate an association between sleep instability and daytime sleepiness [7,8]. The present study suggests that the association between sleep instability and attentional deficits could reflect an impairment of mechanisms regulating both sleep and vigilance, rather than a consequence of daytime sleepiness. Interestingly, our results are compatible with the maturational lag model of ADHD, and could add a developmental perspective of sleep to this model.

## Figures and Tables

**Figure 1 brainsci-11-01425-f001:**
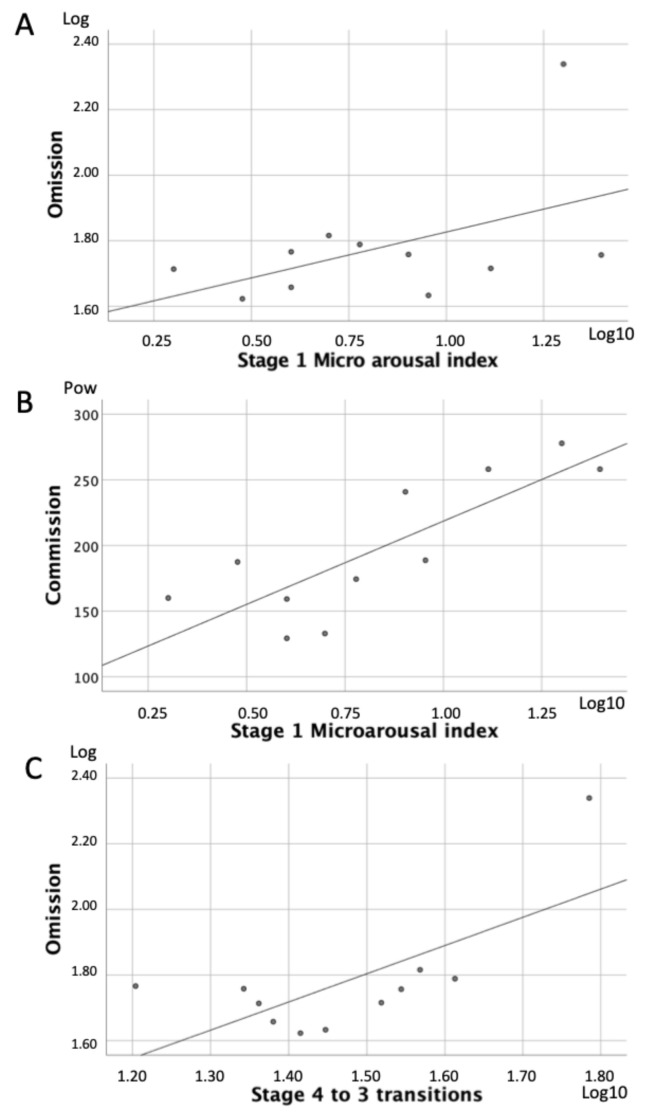
Correlations between sleep stability and attention in children with ADHD. (**A**,**B**) illustrate significant correlations between Stage 1 Microarousal index and attention scores of omission and commission, respectively. (**C**) significant association between omission scores and transitions between sleep stages 4 to 3.

**Table 1 brainsci-11-01425-t001:** Clinical characteristics.

	HC	ADHD	Student-t *p*-Values
Participants (n)	15	12	

Age (years)	10.7 ± 1.6	11.0 ± 1.2	0.620
Body mass index	18.9 ± 5.6	16.7 ± 2.5	0.251
Intellectual Quotient	111.5 ± 10.6	105.5 ± 15.0	0.233

CBCL			
Total score	20.1 ± 15.3	50.3 ± 27.7	0.002 **
Anxiety/Depression score	3.0 ± 3.4	7.2 ± 6.3	0.065
Attention score	2.0 ± 1.8	9.9 ± 4.2	0.000 **

CPT-II			
Omissions	47.3 ± 6.4	67.4 ± 50.3	0.183
Commissions	49.7 ± 12.5	57.8 ± 5.2	0.061
Hit Reaction time	47.9 ± 13.0	51.8 ± 25.7	0.810

CSHQ			
Total score	37.1 ± 0.8	43.1 ± 0.9	0.002 **
Daytime sleepiness	10.9 ± 2.2	12.1 ± 2.6	0.307
Number of children with significant sleepiness	0	1	n.a.

Results are presented as mean ± standard deviation, except for the absolute number of children with significant sleepiness, defined as per the methods section. ADHD: attention deficit hyperactivity disorder; CBCL: Child Behavior Checklist; CPT-II: Continuous performance Test-Second edition; CSHQ: Children Sleep Habits Questionnaire; HC: healthy control; IQ: intelligence quotient; n.a.: statistical analysis not performed. ** *p* < 0.01.

**Table 2 brainsci-11-01425-t002:** Sleep macroarchitecture.

	HC (*n* = 15)	ADHD (*n* = 12)	Student-t *p*-Values
Sleep Latency (min)	15.8 ± 15.4	13.1 ± 11.9	0.517
TST (min)	564.5 ± 77.2	541.0 ± 60.8	0.398
Sleep Efficiency (%)	97.4 ± 2.5	97.4 ± 2.0	0.718
Stage 1 (%)	3.8 ± 1.2	3.2 ± 1.5	0.241
Stage 2 (%)	50.8 ± 4.3	52.1 ± 5.3	0.502
Stage 3 (%)	8.8 ± 2.9	9.7 ± 3.0	0.422
Stage 4 (%)	16.0 ± 4.1	13.7 ± 4.6	0.091
REM sleep (%)	20.5 ± 3.9	21.9 ± 3.9	0.364
REM sleep latency (min)	101.5 ± 43.1	81.1 ± 52.3	0.277
REM sleep duration	132.8 ± 30.2	135.4 ± 30.2	0.820
REM sleep efficiency (%)	87.7 ± 6.8	88.0 ± 4.9	0.909
Mean SpO_2_ (%)	99.3 ± 0.5	99.1 ± 0.6	0.554
Minimum SpO_2_ (%)	81.6 ± 9.3	85.0 ± 5.1	0.309
Time with SpO_2_ < 90% (min)	0.96 ± 1.2	0.51 ± 0.74	0.119
Apnea-Hypopnea Index (no/h)	2.8 ± 1.7	3.1 ± 1.4	0.584
Periodic leg movement index (no/h)	8.7 ± 5.0	10.4 ± 8.0	0.501

Results are presented as mean ± standard deviation. ADHD: attention deficit hyperactivity disorder; HC: healthy control; REM sleep: rapid eye movement sleep; TST: total sleep time; min: minutes; no/h: number per hour of sleep.

**Table 3 brainsci-11-01425-t003:** Sleep stability.

	HC (*n* = 15)	ADHD (*n* = 12)	Student-t *p*-Values
**Awakenings**			
Number	15.7 ± 7.0	15.4 ± 9.9	0.939
Minutes	14.3 ± 12.3	14.6 ± 12.8	0.884
**Microarousal index (no/h)**			
Total	4.6 ± 2.0	5.3 ± 2.0	0.333
Stage 1	12.9 ± 8.0	25.4 ± 19.5	0.034 *
Stage 2	5.9 ± 2.9	6.5 ± 2.6	0.594
Stage 3	6.9 ± 4.3	4.8 ± 3.1	0.159
Stage 4	1.5 ± 1.5	1.8 ± 2.0	0.668
REM sleep	1.2 ± 1.5	2.1 ± 1.9	0.191
**Transitions towards deeper sleep stages**			
Wake to Stage 1	16.7 ± 6.0	16.5 ± 9.4	0.938
Wake to Stage 2	0.7 ± 1.5	0.8 ± 1.2	0.665
Stage 1 to Stage 2	15.2 ± 6.7	14.1 ± 5.9	0.653
Stage 1 to Stage 3	0.1 ± 0.4	0.0 ± 0.0	0.203
Stage 2 to Stage 3	52.7 ± 31.9	43.7 ± 13.3	0.599
Stage 2 to Stage 4	1.1 ± 1.1	0.4 ± 0.5	0.04 *
Stage 3 to Stage 4	20.5 ± 7.3	29.3 ± 13.2	0.064
**Transitions to lighter sleep stages**			
Stage 1 to Wake	5.9 ± 3.4	6.3 ± 4.5	0.937
Stage 2 to Wake	8.7 ± 5.5	8.0 ± 5.2	0.683
Stage 2 to Stage 1	2.9 ± 2.3	2.9 ± 2.6	0.986
Stage 3 to Awake	0.5 ± 0.6	0.2 ± 0.4	0.179
Stage 3 to Stage 2	51.1 ± 31.5	42.8 ± 13.4	0.652
Stage 4 to Stage 2	2.1 ± 1.8	0.9 ± 0.8	0.046 *
Stage 4 to Stage 3	19.6 ± 6.7	28.7 ± 13.2	0.048 *
**REM Sleep Transitions**			
Stage 1 to REM sleep	11.4 ± 6.5	9.0 ± 4.4	0.268
Stage 2 to REM sleep	9.9 ± 2.7	13.1 ± 5.0	0.044 *
Stage 3 to REM sleep	0.3 ± 0.6	0.3 ± 0.5	0.699
Stage 4 to REM sleep	0.0 ± 0.0	0.01 ± 0.3	0.272
REM sleep to Wake	2.4 ± 1.8	2.9 ± 3.1	0.591
REM sleep to Stage 1	12.9 ± 6.3	9.8 ± 5.4	0.191
REM sleep to Stage 2	6.3 ± 2.8	9.5 ± 5.0	0.047 *
REM sleep to Stage 3	0.0 ± 0.0	0.1 ± 0.3	0.272

Results are presented as mean ± standard deviation. ADHD: attention deficit hyperactivity disorder; HC: healthy control; no/h: number per hour of sleep; REM: rapid eye movement. * *p* < 0.05.

## Data Availability

The datasets presented in this article are not readily available, as these data are confidential and require an authorization from the Ethical Review Board of the Hôpital Rivière-des-Prairie to be shared. Requests to access the datasets should be directed to comite.ethique.recherche.cnmtl@ssss.gouv.qc.ca.

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
