# Peer review of "Sleep Instability Correlates with Attentional Impairment in Boys with Attention Deficit Hyperactivity Disorder"

_brainsci, 2021, doi:10.3390/brainsci11111425_

Round 1
Reviewer 1 Report
Dear authors,
thank you for the opportunity to be able to review your research. It was a pleasure to read such a well-written paper. I have only one question/suggestion on how to improve your text.
You are comparing ADHD and HC groups that are highly imbalanced in size. The results of t-test that you are using is well known to be affected by imbalanced group size. There are other statistical tests to use, or ways to prevent this. How did you solve this problem?
I am looking forward to your reply.
Author Response
Indeed, imbalanced group size can lead to unequal variances between samples and affect the type 1 error. Therefore, we used Levene’s test to make sure that the assumption of equal variances was not violated. Our data had non significant Levene’s tests despite the fact that there are 3 more participants in the control group; thus we are comfortable with using t-test. To clarify we add the following sentence to line 194 : “Levene’s test were applied for homogeneity of variance.”
Reviewer 2 Report
The manuscript seems interesting, well written and presented. However, regarding the statistical analysis, I think scoring data that have non-parametric nature should be analysed using non-parametric tests including Mann-Whitney or Wilcoxon matched-pairs signed-rank tests. It would be very pleased to re-evaluate the manuscript after this important issue.
Author Response
We agree with this comment. This is why data without normal distribution were transformed to become normally distributed and meet parametric assumptions. Moreover, Levene’s tests showed that the variances were equal. We thus are comfortable using parametric statistical test and we hope this addresses your concern.